Accuracy of augmented reality navigated surgery for placement of zygomatic implants: a human cadaver study

http://orcid.org/0000-0001-9566-5133 Tabernée Heijtmeijer Sander 1 2 s.j.c.tabernee.heijtmeijer@umcg.nl
Glas Haye 1
Janssen Nard 3
Vosselman Nathalie 1
de Visscher Sebastiaan 1
Spijkervet Fred 1
Raghoebar Gerry 1
http://orcid.org/0000-0001-7128-5814 de Bree Remco 4
Rosenberg Antoine 3
Witjes Max 1
Kraeima Joep 1 2
1 Department of Oral and Maxillofacial Surgery, University Medical Center Groningen , Groningen , Netherlands
2 3D-Lab, University Medical Center Groningen , Groningen , Netherlands
3 Department of Oral and Maxillofacial Surgery & Special Dental Care, Utrecht University Medical Center , Utrecht , Netherlands
4 Department of Head and Neck Surgical Oncology, University Medical Center Utrecht , Utrecht , Netherlands
Tribst João Paulo
Electronic publication date: 2024 Dec 9
Publication date: 2024
Volume: 12
Electronic Location ID: e18468
Received 2024 Jul 1; Accepted 2024 Oct 15
Copyright: © 2024 Tabernée Heijtmeijer et al.
Copyright year: 2024
Copyright holder: Tabernée Heijtmeijer et al.
License: This is an open access article distributed under the terms of the Creative Commons Attribution License, which permits unrestricted use, distribution, reproduction and adaptation in any medium and for any purpose provided that it is properly attributed. For attribution, the original author(s), title, publication source (PeerJ) and either DOI or URL of the article must be cited.
License URL: https://creativecommons.org/licenses/by/4.0/

Keywords: 3D virtual surgical planning, Accuracy, Augmented reality, Guided surgery, Navigation, Zygomatic implants

Funding: Southern Implants (Irene, South Africa) Brainlab (Brainlab AG, Munich, Germany) The zygomatic implants used in this study were sponsored by Southern Implants (Irene, South Africa). The Curve navigation system and associated surgical instruments were made available by Brainlab for the duration of this cadaver study (Brainlab AG, Munich, Germany). The funders had no role in study design, data collection and analysis, decision to publish, or preparation of the manuscript.

==============================
Purpose

Placement of zygomatic implants in the most optimal prosthetic position is considered challenging due to limited bone mass of the zygoma, limited visibility, length of the drilling path and proximity to critical anatomical structures. Augmented reality (AR) navigation can eliminate some of the disadvantages of surgical guides and conventional surgical navigation, while potentially improving accuracy. In this human cadaver study, we evaluated a developed AR navigation approach for placement of zygomatic implants after total maxillectomy.

Methods

The developed AR navigation interface connects a commercial navigation system with the Microsoft HoloLens. AR navigated surgery was performed to place 20 zygomatic implants using five human cadaver skulls after total maxillectomy. To determine accuracy, postoperative scans were virtually matched with preoperative three-dimensional virtual surgical planning, and distances in mm from entry-exit points and angular deviations were calculated as outcome measures. Results were compared with a previously conducted study in which zygomatic implants were positioned with 3D printed surgical guides.

Results

The mean entry point deviation was 2.43 ± 1.33 mm and a 3D angle deviation of 5.80 ± 4.12° (range 1.39–19.16°). The mean exit point deviation was 3.28 mm (±2.17). The abutment height deviation was on average 2.20 ± 1.35 mm. The accuracy of the abutment in the occlusal plane was 4.13 ± 2.53 mm. Surgical guides perform significantly better for the entry-point (P = 0.012) and 3D angle (P = 0.05); however, there is no significant difference in accuracy for the exit-point (P = 0.143) when using 3D printed drill guides or AR navigated surgery.

Conclusion

Despite the higher precision of surgical guides, AR navigation demonstrated acceptable accuracy, with potential for improvement and specialized applications. The study highlights the feasibility of AR navigation for zygomatic implant placement, offering an alternative to conventional methods.

Introduction

‘Portions of this text were previously published as part of a thesis (https://doi.org/10.33612/diss.604892431)’.

Ablative surgery in the maxilla is complicated and is accompanied with profound consequences for the patient’s function and appearance. To improve accuracy of surgical outcome, nowadays the surgeon relies on three-dimensional virtual surgical planning (3D VSP) when bone cuts (osteotomies) are required. 3D VSP typically includes the planning of osteotomies for ablative surgery with a sufficient margin from the tumour borders. Because of the high accuracy of 3D VSP, surgical resections with tumour margin control can be obtained (Kraeima et al., 2018, 2015; Witjes, Schepers & Kraeima, 2018). When one-stage reconstruction surgery is indicated, the 3D VSP also includes the plan for reconstruction. For example, reconstruction and rehabilitation can include location of osteotomies, accurate placement of osteosynthesis materials and dental- or patient-specific implants (Glas, Vosselman & de Visscher, 2020). Traditionally, the 3D VSP is translated towards surgery using surgical guides or image guided therapy techniques like navigation.

Navigation is often used during tumour resection surgery in less accessible locations such as the maxilla and base of the skull. Although reconstruction of maxillary defects with vascularized free flaps appears to yield better speech and swallowing outcomes for extensive defects than conventional prosthetic obturation, not all patients are fit for vascularized free flap reconstruction (Moreno et al., 2010). In these patients, an obturator prosthesis supported by dental or zygomatic implants significantly improves oral function rehabilitation outcomes (Buurman et al., 2020; Schmidt et al., 2004). Placement of dental implants is often not possible due to lack of bone of the maxilla (Jensen, Sindet-Pedersen & Oliver, 1994). Placement of zygomatic implants is considered challenging due to the limited bone mass of the zygoma, limited perioperative visibility, length of the drill path and proximity to anatomical structures like nerve bundles and the orbital cavity (Polido et al., 2023; Al-Nawas et al., 2023). Therefore, guides or navigation are increasingly proposed and investigated for use in zygomatic implant surgery nowadays (Fan et al., 2023; Ramezanzade et al., 2021).

Vrielinck et al. (2003) was one of the first to use drill guides for zygomatic implant surgery. Since then, multiple improvements have been made. Vosselman et al. (2021, 2022) reported on a 3D printed guide using an metal insert for drilling, showing high accuracy in vitro as well as in vivo. One challenge in using 3D printed drill guides is that the bone must be stripped of periosteum to provide stable support for the surgical template. Moreover, 3D printed drill guides can only properly provide control over the entry point of the trajectory, while the deeper trajectory is not as controlled because real time feedback is not possible. Therefore, surgical navigation is considered a promising addition and can even potentially substitute 3D printed drill guided zygomatic implant surgery, mainly due to its real time guidance and feedback.

Extended reality technologies, were first conceived in the 1960s and the first devices became available in the late 1980s. However, augmented reality (AR) was only conceptualized in 1994 and it is only in the last decade that AR has become widely known and increasingly used (Cipresso et al., 2018). AR allows digital information to be overlaid on top of the real world, enabling different types of visualisations, such as projections into the environment, non-in situ visualizations, or projections onto specific related objects, in situ visualizations. Although the use of AR in clinical practice is not yet common, its potential applications are rapidly emerging and hold promise for enhancing medical training, diagnosis, and patient care. AR assisted navigation techniques hold the promise to improve navigation performance during surgery in terms of speed, accuracy and user friendliness (Glas et al., 2021). AR navigation provides in situ visualisation, fusing navigation information directly with the (anatomy of the) patient. Therefore the surgeon does not have to split his attention between the patient and multiple screens in the operating room (Sielhorst, Feuerstein & Navab, 2008). Several AR navigation systems for OMFS have been described, including applications in orthognathic surgery, temporomandibular joint arthrocentesis and dental implantology (Zinser et al., 2013; Tran et al., 2011; Wang et al., 2017, 2019). In this human cadaver study, we use an AR navigation approach to place zygomatic implants after total maxillectomy to assess the accuracy of AR compared to surgical guides.

Materials and Methods

Five formalin fixated human cadaver skulls were obtained and scanned using cone beam computed tomography (CBCT) (ProMax 3D Max; Planmeca, Helsinki, Finland). The settings were in accordance with the clinical settings used for 3D VSP (voxel size 0.4 mm). The scans were converted into a 3D model using ProPlan CMF 3.0 (Materialise, Leuven, Belgium). To mimic the clinical condition and to be able to implant four zygomatic implants per cadaver, a total maxillectomy was included in the 3D VSP. The four zygomatic implants (Southern Implants, Irene, South Africa) were planned towards the most ideal prosthodontic positions based on the pre-maxillectomy situation, in a slightly palatal direction from the occlusal plane. The VSP was performed by an observer with more than 5 years of experience in virtual surgical planning, specialising in oral-maxillo- facial-surgery 3D planning and more than 20 zygoma implants cases. An overview of the 3D VSP can be seen in Fig. 1. The tip of the implant was planned in the lateral cortical bone of the zygomatic complex, with a minimum distance of 2 mm to the orbital cavity. Also, the minimally planned distance between two unilateral implants was 2 mm to ensure sufficient bone around each implant. Hereafter, the 3D models were imported into the navigation software (Brainlab Elements, Brainlab AG, Munich, Germany), in which the drill trajectories for the implants were planned. This study followed the Declaration of Helsinki on medical protocol and ethics. The local Medical Ethics Committee (METc of the University Medical Center Groningen, 2021/504) also granted written permission for the anonymized use of human cadaver data in this study.

Figure 1 Overview of the 3D VSP.

(A) The osteotomy of the maxillectomy is planned. (B) The abutment positions of the zygomatic implants are based on the pre-maxillectomy situation, in a slightly palatal direction from the occlusal plane. (C) The tip of the implant is planned in the lateral cortical bone of the zygomatic complex, with a minimum distance of 2 mm between implants and to the orbital cavity. (D) The 3D VSP is uploaded into the navigation system where the drill trajectories are defined.

In the anatomical lab, the 3D VSP is downloaded on the navigation hardware (Brainlab Curve, Brainlab AG, Munich, Germany), subsequently the patients reference array was attached to the cadaver skulls and registration of imaging data and the cadaver is performed. The registration is performed using a minimum of four preoperatively placed 1.5 mm miniscrews (KLS Martin, Tuttlingen, Germany) which were used as landmarks. A maximum threshold of 1 mm was accepted for the registration error. Hereafter, the surgical drill was calibrated using the instrument calibration matrix (Brainlab AG, Munich, Germany). We used the Microsoft HoloLens I (Microsoft, Redmond WA, United States) as an AR head-mounted-display (HMD). Before the surgical procedure started the HoloLens was connected to the navigation system using an in-house developed workflow, based on previous work described by Glas et al. (2021). This AR visualisation enables the surgeon to visualize and interact with the VSP and navigation data while in the operating room. It collects the VSP from the navigation system and visualises it in the surgical field, while updating the visualisation with real time navigation data. In addition, fiducial markers were added to the HMD using a special made 3D printed reference array. The printed reference array can be seen in Fig. 2. A Brainlab standard four point reference star was selected and the position of the fiducial markers were scanned with a 3D scanner (Artec Spider II, Artec 3D), the HoloLens was scanned as well. Digitally, a design was made with the four markers taking into account the rigidity of the design and user-friendliness, out of the user’s field of vision, but clearly visible to the Brainlab camera during use in the OR. The design was then printed in PA-12 and used for registration of the visualisation, which was done manually. Accuracy of this last step was less important because the information given to user does not depend on this registration. Because of the HoloLens reference array, the position of the surgeon is tracked by the navigation system as well, enabling a semi-automated registration of the virtual overlay onto the patient and surgical tools. A quick manual registration had to be performed before each use, when the HMD was put on. The virtual overlay enabled visualizing directions of the planned trajectories and navigation directions of the instruments, real live in the surgical field. An example of the AR navigation setup is seen in Fig. 3.

Figure 2 The digital model of the designed reference array for the HoloLens, to track the position of the user.

The digital model of the designed reference array for the HoloLens is shown. It consists of a model of a standard 4-point reference array that can be used with the brainlab system and a holder to attach the reference star to the HoloLens. The model sits rigidly and always in the same place due to its design and snap hooks. For the experiment, it was printed in PA-12 using an SLS 3D printing method.

Figure 3 AR navigation system and interface.

(A) Image of the setup during the cadaver experiment. The surgeon wearing the HoloLens with a custom made reference array attached. Fiducial markers on the reference arrays of the HoloLens, surgical drill and skull enable the navigation system to track the objects and project the visualisation on the patient. (B) An example of the AR visualisation as seen by the surgeon during the phantom setup. Virtual navigational indicators are projected into the surgical area. The colour and size of the green circles change with manipulating the direction of the drill. The depth to target is also updated real time.

Tasks and participants

The surgery was performed by experienced (>10 years) OMF surgeons skilled in 3D VSP and navigated surgery. Three OMF surgeons (NJ, SV, GR) drilled the implant trajectories and placed the zygomatic implants. All but one of the surgeons participated in a training session where dental implants were placed in sawbones using the same workflow and AR visualization. After the cranial resection margin of the maxilla was marked using the AR navigation, a total maxillectomy was performed. Hereafter, a total of 20 zygomatic implants were placed in five cadavers using the AR navigation. No sample size was calculated for this study, as it is a pilot investigation and there was no prior data available on the system’s accuracy. Additionally, accuracy can vary significantly depending on the user, the specific system, and the application, making it difficult to base a sample size on previous studies. Therefore, it was not feasible to perform a formal sample size calculation for this pilot.

Implant placement accuracy

For determining the zygomatic implant accuracy, a postoperative CBCT scan was made. A 3D model was made in a similar fashion as for the 3D VSP. Hereafter the postoperative 3D skull was matched virtually with the preoperative 3D VSP using an ICP algorithm and visually confirmed (Besl & McKay, 1992). Subsequently, bone entry and exit positions in the zygomatic bone were defined by the intersection of the long axis of the implant with the bone. Accuracy was assessed in an identical way to the method used by Vosselman et al. (2021, 2022) where two coordinate systems have been used. Both coordinate systems are illustrated in Fig. 4 and defined as:

1) The implant’s coordinate system (ICoS): the z-axis runs along the long axis of each planned implant.

2) The occlusion coordinate system (OcoS): congruent with the axial, sagittal and coronal planes. The axial plane is defined by the occlusion plane derived from the positions of the planned abutments.

Figure 4 Reference planes and coordinate systems.

Reference planes and coordinate systems for assessing the accuracy of zygomatic implant. In red the planned zygomatic implant position, in gray the postoperative zygomatic implant position derived from CBCT. (A) The implant coordinate system (IcoS) including the three reproducible reference planes in which the accuracy is measured; the centre of the abutment, the bone entry point, and bone exit point of the implant. (B) Visualisation of the occlusion plane coordinate system (OcoS). The occlusal plane is defined parallel to a plane of the occlusion, the ideal prosthetic occlusional plane. Perpendicular to this plane is the blue arrow which indicates the direction the abutment height accuracy is calculated.

Accuracy of the abutment, the entry and exit points were measured in the IcoS. The distance between planned and postoperative position was measured in a plane perpendicular to the long axis of the implant. Other accuracies were measured in the OcoS, these include the height deviation of the abutment in the occlusal plane, the displacement of the abutment in the occlusal plane, the axial, coronal, sagittal and 3D angle. A placement accuracy of 3 mm in the occlusal plane for the abutment was considered to be successful, and is assumed to result in a passive fit for a prosthesis (Vosselman et al., 2021, 2022). The accuracy was compared to the accuracy of guided placement, on which we have reported earlier (Vosselman et al., 2021). This study was performed in a similar fashion, the VSP was followed by guided placement of 10 zygomatic implants using five fresh frozen human cadavers. The postoperative analysis, based on the postoperatively performed CBCT, was performed identically to that study.

Results

With the aid of the VSP, the navigated drill and AR navigation a total 4 maxillectomies were performed and 20 zygomatic implants were placed in five cadaver heads. In one of the cadavers, a maxillectomy had previously been performed. This maxillectomy has been digitally copied preoperatively into the 3D VSP.

The implant lengths varied between 40 and 55 mm. The mean entry point deviation was 2.43 ± 1.33 mm and a 3D angle deviation of 5.80 ± 4.12° (range 1.39–19.16°). The mean exit point deviation was 3.28 mm (±2.17), and the abutment height deviation was on average 2.20 ± 1.35 mm. The accuracy of the abutment in the occlusal plane was 4.13 ± 2.53 mm. The complete accuracy results can be seen in Table 1. No significant differences were found between ventral and dorsal implants (P > 0.05) nor between left or right implants (P > 0.05).

Table 1 Accuracy data.

	Augmented Reality	Guides (Vosselman et al., 2021)		
		Mean (±SD)	Range	Mean (±SD)	Range	P value	
ICoS deviation	Abutment (mm)	3.34 (±2.11)	0.60–8.63	1.19 (±0.63)	0.53–3.42	0.005	
Entry-point (mm)	2.43 (±1.33)	0.60–5.96	1.20 (±0.61)	0.43–3.24	0.012	
Exit-point (mm)	3.28 (±2.17)	1.36–11.65	2.12 (±1.24)	1.11–4.72	0.143	
OCos deviation	Abutment in occlusal plane (mm)	4.13 (±2.53)	1.09–2.53	1.77 (±1.31)	0.87–6.04	0.012	
Abutment height from occlusal plane (mm)	2.20 (±1.35)	0.08–4.63	1.03 (±0.85)	0.01–6.58	0.021	
Axial angle (°)	5.76 (±4.74)	1.04–21.63	2.07 (±2.63)	0.19–4.34	0.062	
Coronal angle (°)	2.44 (±2.02)	0.27–7.63	0.99 (±2.32)	0.25–7.97	0.682	
Sagittal angle (°)	7.62 (±6.55)	0.39–25.27	1.48 (±3.59)	0.27–7.04	0.047	
3D angle (°)	5.80 (±4.12)	1.39–19.16	2.97 (±1.43)	0.34–6.13	0.051	
Notes:

Results of the postoperative analysis of the implant coordinate system (IcoS) and occlusion coordinate system (OcoS) measurements for the AR system (this article) and with the use of patient-specific guides (previous work) (Vosselman et al., 2021). Statistical significant differences are highlighted in bold.

Compared to the results of our previous study on the accuracy of zygomatic implants with 3D-printed surgical guides (Vosselman et al., 2021), all three accuracy measurements of the abutment and the entry-point were significantly more accurate with the use of surgical guides. For the exit-point there was no statistical significant difference. All results of the AR navigation (this study) vs. guides (previous work) are found in Table 1.

Discussion

This study shows a novel AR dynamic navigation system for placement of zygomatic implants. Using AR navigation 20 zygomatic implants have been placed in five cadavers. Surgical guides perform significantly better for the entry-point (1.20 ± 0.61 mm vs. 2.43 ± 1.33 mm), however no statistical differences could be found for the exit-point or 3D angle. Placement of zygomatic implants at time of ablative surgery has been shown to be an effective means of accelerating oral function rehabilitation, along with early loading protocols. Placement of zygomatic implants is challenging, due to the length of zygomatic implants (40 to 55 mm), a minor angular deviation can lead to relatively large positional errors at the exit point.

Navigated zygomatic implant placement was described before in multiple studies (Wang et al., 2018; Pellegrino et al., 2015; Gasparini et al., 2017; Pellegrino et al., 2020; Fan et al., 2023). Some studies use clinically available navigation systems (Wang et al., 2018; Hung et al., 2017; Hung et al., 2016), some used dental navigation systems or developed their own (Pellegrino et al., 2015; Gasparini et al., 2017; González Rueda et al., 2022; Kreissl et al., 2007; Panchal et al., 2019; González-Rueda et al., 2023), extended reality systems are also described (González-Rueda et al., 2023; Fan et al., 2024). Zhou et al. (2021) report on 14 navigated placed zygomatic implants in patients with maxillectomy defects, with an mean accuracy at the entry-point of 1.56 ± 0.54 mm, exit-point of 1.87 ± 0.63 mm (exit point), and an angle deviation of 2.52 ± 0.84°. Chrcanovic, Oliveira & Custódio (2010) placed 16 zygomatic implants in human cadavers with an angle accuracy of 8.06 ± 6.40° for the anterior-posterior view and 11.20 ± 9.75° for the caudal cranial view. Vrielinck et al. (2003) report on an entry-point accuracy of 2.77 mm (range 1.0–7.4) and exit-point accuracy of 4.46 mm (range 0.3–9.7) in a patient cohort. In our study the accuracy of implant placement in human cadavers after total maxillectomy was slightly higher both for the surgical guides from our previous study with two of the same surgeons (SV, GR) and the AR navigation (Vosselman et al., 2021). Hung et al. (2017) have placed 40 zygomatic implants in severe atrophic maxillae using the Brainlab navigation system (Brainlab AG, Munich, Germany), reporting an accuracy of 1.35 ± 0.75 mm (entry-point), 2.15 ± 0.95 mm (exit-point), and 2.05 ± 1.02° angle deviation. However, accuracy of zygomatic implantation in a resorbed maxilla might be higher due to a more stable drill entry point. While drilling the trajectory after (total) maxillectomy, the bone entry location is not a stable flat surface. Most likely the tip of the drill approaches the anterior wall of the maxillary sinus in an oblique fashion, making an exact entry-point difficult due to sliding of the drill tip along this cortical bone. Using navigation guidance, the surgeon is likely to correct for this entry-point deviation by manipulating the direction of the drill, in order to get back on the planned trajectory. One observation we made, is that manipulating the drill could subsequently cause the drill to bend, mainly due to the length of the drill bits. As a result, the tip of the drill no longer matches the virtual drill in the navigation, which in turn leads to additional inaccuracy. In none of the five cases the placement of the implants was complicated by an orbital perforation. Therefore, accuracy in this small sample seems to be accurate enough for safe application in human patients of both investigated methods; drilling guides as well as AR navigation. Additionally, the abutment height was within the 3 mm limit for both surgical guides as well as for AR navigation. Positioning of the abutment is important for immediate placement of implant-retained obturator prosthesis (Vosselman et al., 2022), this is possible when the placement is within this 3 mm limit. Larger deviations may result in prosthetic rehabilitation not being possible immediately, or not being possible at all with the implant positioning. The entry and exit point are important for stable implant placement and long-term success (Şahin et al., 2024). Deviation from these can additionally cause immediate complications when crucial structures, nerve bundles and the orbital cavity are hit. Deviations from the entry and exit points are less critical as long as stable positioning in the bone is maintained and vital structures are avoided. All implants were stable, however, as this is a cadaveric study, long-term success could not be assessed. Thus, despite the difference in accuracy, there is no difference in clinical outcome between the AR navigation and guided positioning. Multiple factors impact the accuracy of surgical navigation, including imaging techniques, registrations procedures of the patient as well as the surgical tools, how rigid these surgical tools are, and moreover the human machine interface which influences the surgeon’s performance.

In future research, the use of a stiffer drill bit or drill bus could enhance accuracy during drilling and the initial calibration process. Additionally, in this study, registration was performed using four markers placed via screws in the skull—a method feasible in cadaveric studies but not commonly used in patients, where soft tissue registration is commonly used. However, soft tissue registration is not viable in cadavers due to preservation in formalin (Grauvogel et al., 2010; Taleb et al., 2023). For improved accuracy in future studies, registration using a 3D CBCT scan could offer a more precise alternative, allowing for better alignment and positioning during surgery, which is also used for patients (Mischkowski et al., 2007). Surgeon experience also plays a major role in the final accuracy of placement, as the implants with the AR navigation system and guides were placed by the same surgeons, we have tried to limit this confounder. Using surgical guides, an accurate result can be obtained, however, sufficient bone has to be exposed for the guide to be stable during the entire drilling procedure. Moreover, the guide has to be designed such that it minimises the risk of deforming during drilling. This means sufficient support area and sometimes a bulkier guide. In ablative oncological surgery, the surgical area might be more easily accessible while this is more restricted in elective procedures. When a minimal invasive procedure is warranted AR navigation could be used as an alternative. However, based on the results of this cadaver study probably the best results may be obtained, if surgical guides are used for control of the entry point and AR navigation for trajectory control.

A limitation of this study is that an inter-observer variability for accuracy of the measurements was not assessed. However, the same technique was used as in our previous study (Vosselman et al., 2022) and the measurements were conducted by the same observer. The intraclass correlation coefficient determined in that study was excellent for the abutment position, entry-point, exit-point, and 3D angle respectively 0.91 mm, 1.00 mm, 0.97 mm, and 0.98°. Therefore it was deemed unnecessary to repeat inter-observer variability analysis. The assessment did not include an inferiority analysis; thus, while guides were not significantly more accurate in all measurements and therefore superior to the AR system, it cannot be ruled out the possibility that the performance of the AR system could be inferior. This limitation underscores the need for further research that specifically investigates inferiority to provide a more comprehensive understanding of the best approach for positioning of zygomatic implants. Recently, the use of task-autonomous robots in zygomatic implant placement has attracted attention and shown promising results alongside traditional static guides and dynamic navigation systems (Deng et al., 2023). A key advantage of robotic systems is their ability to operate with high precision and consistency, minimising human error and increasing surgical efficiency. However, a major limitation is that robots cannot assess bone density and cannot easily adjust their plans during surgery, making them less flexible (Mai, Dam & Lee, 2023). Moreover, factors such as high costs, the need for specialised training and the increased complexity of surgical procedures can make their implementation challenging. In contrast, AR navigation provides real-time visual guidance, allowing surgeons to remain aware of the situation and make immediate adjustments as needed, although further research should determine the best approach for each case.

While this study focused on AR navigation in one stage resection and reconstruction surgery for placement of zygomatic implants, the AR-navigation might also be used in other craniomaxillofacial indications such as maxilla, orbital, and cranial base resections.

Conclusions

This study shows a novel AR dynamic navigation system for placement of zygomatic implants. Despite the fact that patient specific guides lead to a more accurate placement compared to AR navigation, the accuracy of AR navigation is acceptable as well and the authors are convinced that it will continue to improve and will find its specific application. The study highlights the feasibility of AR navigation for zygomatic implant placement, offering an alternative to conventional methods.

Supplemental Information

Supplemental Information 1 Raw data.

Accuracy Measurements for the implant accuracy of the zygomatic implants per implant per cadaver.

Additional Information and Declarations

Competing Interests

Author Contributions

Human Ethics

Data Availability

The authors declare that they have no competing interests.

Sander Tabernée Heijtmeijer analyzed the data, prepared figures and/or tables, authored or reviewed drafts of the article, and approved the final draft.

Haye Glas conceived and designed the experiments, performed the experiments, analyzed the data, prepared figures and/or tables, authored or reviewed drafts of the article, and approved the final draft.

Nard Janssen conceived and designed the experiments, performed the experiments, authored or reviewed drafts of the article, and approved the final draft.

Nathalie Vosselman conceived and designed the experiments, performed the experiments, analyzed the data, authored or reviewed drafts of the article, and approved the final draft.

Sebastiaan de Visscher performed the experiments, authored or reviewed drafts of the article, and approved the final draft.

Fred Spijkervet conceived and designed the experiments, authored or reviewed drafts of the article, and approved the final draft.

Gerry Raghoebar conceived and designed the experiments, performed the experiments, analyzed the data, authored or reviewed drafts of the article, and approved the final draft.

Remco de Bree conceived and designed the experiments, authored or reviewed drafts of the article, and approved the final draft.

Antoine Rosenberg conceived and designed the experiments, performed the experiments, authored or reviewed drafts of the article, and approved the final draft.

Max Witjes conceived and designed the experiments, performed the experiments, analyzed the data, authored or reviewed drafts of the article, and approved the final draft.

Joep Kraeima conceived and designed the experiments, analyzed the data, authored or reviewed drafts of the article, and approved the final draft.

The following information was supplied relating to ethical approvals (i.e., approving body and any reference numbers):

Medical Ethics Review Board of the University Medical Center Groningen (METc Number: 2021/504)

The following information was supplied regarding data availability:

The specimen images are available at MorphoSource:

- Media 000678970: Frontal Part Of Skull, DOI 10.17602/M2/M678970.

- Media 000678975: Frontal Part Of Skull, DOI 10.17602/M2/M678975.

- Media 000678980: Frontal Part Of Skull, DOI 10.17602/M2/M678980.

- Media 000678985: Frontal Part Of Skull, DOI 10.17602/M2/M678985.

- Media 000678990: Frontal Part Of Skull, DOI 10.17602/M2/M678990.

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
