# Peer review of "Accuracy of augmented reality navigated surgery for placement of zygomatic implants: a human cadaver study"

_PeerJ, doi:10.7717/peerj.18468_

## Round 0.1 · original submission · Minor Revisions

Dear author,

Thank you for submitting your manuscript "Accuracy of augmented reality navigated surgery for placement of zygomatic implants; a human cadaver study" to PeerJ. We request minor revisions, including clarification on the surgeon's experience to address potential heterogeneity, additional accuracy metrics (e.g., fiducial marker accuracy), and further discussion of the clinical significance of deviations. The exit-point accuracy findings should be explored, and the measurement protocol clarified, ideally with two independent assessors. Additionally, the limitations require further elaboration, the discussion should include the clinical impact of accuracy on prosthesis outcomes, and the results should be presented using both Violin and Box Plots. I look forward to your revised submission.

Kind regards,
Dr. Tribst JPM
ACTA

Reviewer 1 ·

Basic reporting

This cadaver study evaluated a developed AR navigation approach for the placement of zygomatic implants after total maxillectomy. The topic is fascinating and overall well written but needs some revision.
Your introduction needs more detail. I suggest that you add the description of the development and current state of AR(e.g. AR visualization is divided into in situ visualization and non-in situ visualization) in lines 83- 86 to provide more justification for your study.

Experimental design

As stated in your introduction, AR navigation is an improved version of dynamic navigation, so why did you set the control group as a surgical guide instead of the navigation? I suggest you compare the results with the previous study guided by dynamic navigation or add a control group of dynamic navigation-guided zygomatic implant placement.
Digitally guided zygomatic implant surgery relies heavily on the experience of the operating surgeon, and whether the surgeon in this study was in the same group as the surgeon previously involved in the zygomatic implant placement guided by the surgical guides, and if not, the data from the two groups has a large degree of heterogeneity, and the results of the comparison are not scientifically sound.

Validity of the findings

Currently, in addition to static guides and dynamic navigation, there are also studies applying task-autonomous robots in zygomatic implant placement with favorable results. Do you think AR navigation has any advantages and disadvantages compared to the task-autonomous robot? Please state them in the discussion.

Additional comments

1.Please provide the sample size calculation process.
2.Lines 119-120: The combination of AR and navigation is a great idea, but the description of the method is too brief, please explain the method with detailed images and text.
3.Lines 124-125: This looks like a great approach, please elaborate on the design and distribution of HMD's fiducial markers, and the registration process. Preferably with illustrations.
4.Lines 149-153: Please show each measurement in a schematic diagram.
5.Line 182: It may be possible to demonstrate the differences between the two studies more visually with a box plot diagram.

·

Basic reporting

Thank you for submitting such an interesting research manuscript. Below are my comments:
Strengths:
1. Relevance and Importance:
o The study addresses a clinically relevant issue: the challenging placement of zygomatic implants due to limited bone mass, visibility, and proximity to critical structures.
o Augmented reality (AR) navigation is an emerging technology, and its potential benefits are explored. The authors also further compared the results of this study with their previous published work using surgical guides, to show the difference in accuracy between AR and surgical guides.

2. Novelty:
o The use of AR navigation for zygomatic implant placement after total maxillectomy is innovative.
o The study contributes to the field by evaluating a new approach, as the authors suggested, combining both surgical guide and AR navigation could improve further the accuracy and predictability of the surgery.
o The English language used throughout the manuscript was of high proficiency level, making the reading of this manuscript a pleasant experience, though it is advisable for the authors to provide the raw data of previous work using surgical guides in Excel file for this submission too, to allow the reviewers to have a thorough assessment and analysis of the raw data and results of their previous work on surgical guides, since the authors were making comparison between the AR study and surgical guide study as part of this current submission.

Experimental design

Strength
Methodology:
o The study uses human cadaver skulls, providing a realistic anatomical context.
o Virtual surgical planning and postoperative scans enhance accuracy assessment.
o Comparison with 3D printed surgical guides adds valuable context (though it must be specified clearly in the "Methodology" as well in the column of "Table 1" that the "Guide" data was retrieved from another research work and not the current work), to provide clarity to the readers too.
* * *
Weaknesses:
1. Sample Size:
o The sample size (20 zygomatic implants) may be small for drawing definitive conclusions. Hence the reviewer's question in the "Comments" section of the reviewed version of the manuscript on "Is the sample size calculated?". The reviewer is also wondering if the sample size was calculated or not for the authors' previous work on surgical guide using 10 zygomatic implants.
o Larger sample sizes would strengthen the study’s findings.

2. Precision Metrics:
o While entry point deviation and 3D angle deviation are reported, additional metrics accuracy assessment (e.g., the accuracy of the fiducial markers added to the HMD) could provide a more comprehensive assessment.
o The clinical significance of these deviations should be discussed.
3. Exit-Point Accuracy:
o The lack of significant difference in exit-point accuracy between 3D printed guides and AR navigation, despite such a wide range of deviation, warrants further exploration.
o Discuss potential reasons for this finding (e.g., limitations of AR visualization).

3. Calibration of Surgeons:
o Noted one of the three surgeons undergone the training using sawbones. However, there was no explicit statement in the manuscript if all the three surgeons involved in performing the surgery were calibrated their navigated surgical skills or not, as this could be another factor that might introduce heterogeneity in the research results.

Validity of the findings

1. Please provide the raw data of the previous work using surgical guides in Excel file to allow for thorough review of the data presented in "Table 1" of this current submission.
2. Please also provide the sample size calculation of this current submission as well as the previous work using surgical guide as a separate file, if there is any.

Additional comments

1. Discussion of Clinical Relevance:
o Elaborate on how the reported deviations (entry, exit, abutment height) impact clinical outcomes.


2. Potential Sources of Error:
o Address potential sources of error related to AR navigation (e.g., registration accuracy, surgeon’s proficiency in navigated surgery and calibration before the commencement of research).
o Discuss strategies to minimize these errors in clinical practice.

3. Future Directions:
o The authors suggested that both surgical guide and AR could be combined to increase both the entry point and trajectory accuracy, which is interesting and they should explore this aspect too in coming research
o Explore specialized applications where AR navigation excels (e.g., complex cases, revision surgeries).

Reviewer 3 ·

Basic reporting

I would like to thank the editorial team and the authors for the opportunity to review the manuscript entitled "Accuracy of augmented reality navigated surgery for placement of zygomatic implants; a human cadaver study".

Overall, a very interesting paper. The authors placed 20 zygomatic implants in 5 human cadavers using an AR-navigated method. This was then evaluated in a postoperative CBCT scan. The results were compared with a previous study (similar setting?). The limitations need to be better highlighted. Otherwise, I have some minor comments before the paper can be published.

Major points:
- The authors did not perform an inferiority study, they only tested for superiority.
Inferiority studies are based on analysis with confidence intervals. Their study data can be the basis for an inferiority study. The authors should weaken the conclusion of their study to the effect that inferiority cannot be excluded. This can also be done in the limitation.
- How were the implants measured? By one person only? If so, you should have it done by at least 2 people independently and calculate the mean of both investigators. Furthermore, you should provide the exact calculation formulas so that it is really clear what you have actually measured.
- The limitations should be worked out more.
- The discussion is too short, it should also have more discussion regarding the clinical context, especially what high or low accuracy has an impact on the final prosthesis supply.
- In addition, the results should be presented with a Violin Plot and a Box Plot.

Minor points:
- Lines 68-73 should have a source.
- What was the surgical experience of the person who planned the implants? Authors should include this information in the Materials and Methods.
- "granted written permission for the retrospective anonymized use of human cadaver data in this study". This is not a retrospective study. The authors should correct this sentence.
- In Figure 2a, the authors should blur the eye area for anonymization purposes.
- "Glas et al[11]", it should be et al. (et alter)
- What was the exact background of the surgeons? How many zygomatic implants had they previously placed in real patients and/or phantom models?
- What was the registration method? PPR, ICP, IBR?

Experimental design

The authors placed 20 zygomatic implants in 5 human cadavers using an AR-navigated method. This was then evaluated in a postoperative CBCT scan. The results were compared with a previous study (similar setting?).

Validity of the findings

Valid, the measurements in the CBCT should be carried out by 2 independent persons.

Additional comments

None

---

## Round 0.2 · accepted · Accept

Dear author,

Upon reviewing the revised version of your manuscript, I confirm that all of the reviewers' comments have been addressed satisfactorily. I have carefully reviewed the changes internally and are happy with the current version of your submission.

The manuscript is now ready for publication in PeerJ. You will be notified shortly regarding the next steps in the publication process.

I look forward to your continued contributions!

Kind regards,
Dr. Tribst JPM
Academisch Centrum Tandheelkunde Amsterdam

Reviewer 1 ·

Basic reporting

no comment

Experimental design

If calculating the sample size from previous research data is not feasible, calculating it based on the effective size may be possible.

Validity of the findings

no comment

·

Basic reporting

I am satisfied with the corrections done by the authors.

Experimental design

The authors had clarified that their study is a pilot study explicitly in their revised manuscript, which had addressed my previous comments and suggestions succinctly.

Validity of the findings

I am satisfied with all the rebuttal comments and corrections done by the authors.

Additional comments

Congratulations on having executed and written such an interesting research manuscripts!

Reviewer 3 ·

Basic reporting

All my points have been answered satisfactorily. I recommend to accept the manuscript.

Experimental design

All my points have been answered satisfactorily. I recommend to accept the manuscript.

Validity of the findings

All my points have been answered satisfactorily. I recommend to accept the manuscript.

Additional comments

All my points have been answered satisfactorily. I recommend to accept the manuscript.